# Perception of physicians towards electronic prescription system and associated factors at resource limited setting 2021: Cross sectional study

**Gizaw Hailiye Teferi** [1]*, **Tewodros Eshete Wonde**[1‡], **Maru Meseret Tadele**[1‡], **Bayou Tilahun Assaye**[1‡], **Zegeye Regasa Hordofa**[1‡], **Mohammedjud Hassen Ahmed**[2‡], **Samuel Hailegebrael**[3‡]

**1** Department of Health Informatics College of Health Science Debre Markos University, Debre Markos, Ethiopia, **2** Department of Health informatics College of Health Science Mettu University, Mettu, Ethiopia, **3** Department of Health Informatics College of Medicine and Health Science Arba Minch University, Arba Minch, Ethiopia

‡ TEW, MMT, BTA, ZRH, MHA and SH are second equal contribution to this work.
* ghailiye463@gmail.com

**Data Availability Statement:** All relevant data are within the paper and its Supporting information files.

## Abstract

### Introduction

The healthcare industry is increasingly concerned about medical errors, which are the leading cause of death worldwide and also compromise patient safety. This medical error is even more serious in developing countries where healthcare is not supported by technology. Because of the traditional paper-based prescription system, Ethiopia has an overall medication prescribing error rate of 58.07% that could be avoided if an electronic prescription system was in place. Therefore, this study aims to assess physicians' perceptions towards electronic prescription implementation.

### Methods

From February 1 to April 5, 2021, an institution-based cross-sectional study was conducted among physicians working in public hospitals in the Amhara region. 384 physicians were selected using a simple random sampling method. The data was collected using a self-administered questionnaire and analysed using SPSS, version 21. To assess factors associated with perception among physicians, a binary and multivariable logistic regression analysis were performed. A P.05 value, at a 95% confidence interval, was considered statistically significant. The validity of the questionnaire was determined based on expert opinion, as well as its reliability was determined by calculating the value of Cronbach alpha (α = .78).

### Results

In this study, 231 (76.5%) of study participants had a positive perception of electronic prescription. Around 70.8% had more than 5 years of computer usage experience. Nearly 90%

**Funding:** The author(s) received no specific funding for this work.

**Competing interests:** The authors have declared that no competing interests exist.

**Abbreviations:** AOR, Adjusted odds ratio; e-Prescription, Electronic prescription; m-Health, Mobile Health; SPSS, Statistical package for social science.

of participants claimed that their prescriptions were legible; however, 89% believe that paper-based prescriptions are prone to error. According to multivariable logistic regression analysis, technical skill [AOR] 4.7, 95% confidence interval [CI] (1.27–17.41), good internet access (AOR 2.82, % CI 1.75–4.54), and perceived usefulness of e-prescription system (AOR 3.31, 95% CI 1.01–12.12) were significantly associated with perception.

## Conclusions

The majority of respondents have a positive perception of electronic prescription. The most notable factors associated with physician perception were organizational factors, internet access, perceived usefulness of the system, and technical skill.

## Introduction

The health care industry has become increasingly concerned about patient safety, which corresponds to societal trends [1]. Medication safety is one of the most important concerning issues in global health policy [2]. Medical error is a common encounter and represents an important public health problem posing a serious threat to patient safety. A medical error is an error occurred in the prescribing, dispensing, or administration of a drug [3]. It is an avoidable negative impact of medical care, regardless of whether it is a visible or detrimental to the patient [4]. According to a study released in 2016, medical error is the third leading cause of death in the United States, following Cardio disease and cancer [5]. Researchers examined studies that analysed medical death rate data from 2000 to 2008 and extrapolated that over 250,000 deaths per year were caused by a medical error, which contribute to 9.5% of all deaths in the United States of America each year [5].

Across the globe, medication errors are the leading causes of avoidable patient harm in the health care system [4]. In African health care, setting medication errors are common health problems [6].

According to a systematic review and meta-analysis study conducted in Ethiopia, the overall medication error rate was found to be 57.6%, with the magnitude of medication administration and prescription error being 58.4% [7].

A study conducted in the northeast part of Ethiopia (Dessie referral hospital) found an overall medication prescribing error rate of 58.07%, with incomplete prescriptions and dosing errors being the most common error types that could be avoided if an electronic prescription (e-prescription) system was in place [8]. Paper-based prescriptions have long been the preferred method of communication for physicians making medication therapy decisions and pharmacists dispensing medications. It is also recognized as a valuable resource for patients in terms of how to use the medicine to achieve the greatest benefit [9]. However, literature revealed that e-prescription is more reliable when it comes to patient safety than the traditional paper-based prescription method [10].

The use of an e-prescriptions has the potential to improve the quality of patient care at the pharmacy [1]. Indeed, a study shows a significant improvements associated with an e-prescription system implementation, including a 86% decrease in serious medication errors, and an increase in Medicare formulary adherence from 14% to 88% [11]. e-Prescription is an alternative to many years old paper-based prescriptions. Electronic prescribing and dispensing processes of drugs whether in medical practice, follow up or research has become an integral part of pharmacy informatics [12]. Despite the fact that the e-Prescription system is an essential

tool for the healthcare industry, e-Prescription adoption and utilization remain low in developing countries [13]. Many healthcare organizations around the world have implemented electronic information systems to improve the process of recording information, but only a few have succeeded [14].

The percentages of failure to achieve the desired benefits from the implementation of an electronic information system are alarmingly higher. Globally, more than half of electronic information system projects, including e-Prescription, failed before they reached their goals [15]. Numerous reasons are given for low electronic system adoption such as resistance by the users, the opposition of transition from paper-based to the electronic system, technical competency of the frontline users, and ease of use of a given electronic system, lack of organizational readiness, and security and confidentiality issue [15–19].

One of the major contributing factors in the successful implementation of e-Prescription is health care professionals' technical competency [17, 19]. In addition to the physician competency to use e-prescription error, free e-prescription requires regular technical assistance [17, 19, 20].

The infrastructure of a given health care organization and the culture of the society in which the organization operates govern the readiness of the professional to implement e-prescription [21]. In addition to this, the overall readiness of healthcare organizations to implement e-prescription is affected by individual-related factors [22]. According to the Technology Acceptance model ease of use of a given information system influences users' perception and outcomes [17, 23]. The other factor that governs the perception of the physician to use an e-Prescription systems is the security concern of electronic systems. Electronically transmitted data could be attacked and be used for other criminal purposes such as inappropriate prescribing of controlled substances or high-cost medications [24]. Despite the fact that most e-prescribing systems have measures in place to secure access, they are vulnerable to hacking.

As successful adoption of electronic prescription system or any electronic health information system is a change process it needs many behavioural modifications in the work environment for physicians [25].

Perception assessment, as a comprehensive measure to provide a proper image of existing conditions and the preparedness of healthcare organizations to change, is also a way to identify potential causes of failure in innovation [26]. This study, therefore, aims to assess the perception of physicians towards electronic prescription system implementation and associated factors at public hospitals in East and West Gojjam zones North Ethiopia.

## Method

### Study design and setting

This was a cross-sectional, questionnaire-based study done to assess physicians' perception towards implementation of an electronic prescription system and associated factors at public health hospitals in East and West Gojjam zones north Ethiopia 300km from Addis Ababa. The sample size was computed as 384 which was 80% of the total physicians in the study setting during the data collection period. Using single population proportion formula taking 50% at 95% confidence level assuming a 5% margin of error.

$$\text{Sample size (n)} = \frac{\left(Z^{\alpha}/2\right)^2 x\, p(1-P)}{d^2}, \; \text{(n)} = \frac{(1.96)^2 x\, 0.5(1-0.5)}{(0.05)^2} = 384.2$$

Where;

- n = estimated sample size

- p = single population proportion (50%).

- $Z_{\alpha/2}$ = is value of standard normal distribution (Z-statistic) at the 95% confidence level ($\alpha$ = 0.05) which is 1.96,

- d = is the margin of error 5% (0.05)

There are 17 hospitals in the west and east Gojjam zones and all the 17 hospitals were taken and then proportional allocation was made for each hospital in order to assure representativeness of the sample. Finally simple random sampling method was used. 384 participants were recruited using the lottery method (each member of the population was assigned a unique number and the number was written on separate paper/card/ with the same size, the card mixed in the basket well and sample drown) and formed the sample. The survey consisted of 35 questions encompassing the following domains: 1. sociodemographic characteristics, 2. Current prescription status, 3. Current computer usage status 4. Physicians' technical skill, 5. Perceived usefulness, and 7. Organizational factors.

A self-administered questionnaire was adapted from the previous studies [17, 27]. To ensure the validity of the questionnaire, an expert panel (10 doctors having at least 5 years' experience in general practice or primary care research) with different specializations were invited to review the tool and revise it and from pilot study reliability was calculated result was $\alpha$ = 0.78.

Before the actual data collection, pilot testing of the questionnaire was conducted among 20 physicians at Debre tabor hospital to check internal consistency within the questioners. Then necessary correction was done based on the pre-test finding.

Two-days training was given by the principal investigator for (five data collectors) on the objective of the study and data collection procedures. Data were collected from February 5 to March 30, 2021, using a self-administered questionnaire, one data collector was assigned for each hospital and the supervisor facilitated the data collection process. The principal investigator and supervisors did daily supportive supervision on data collectors. Data backup activities, like storing data at different places and putting data in different formats (hard and soft copies) were performed to prevent data loss.

## Study variables and operational definitions

**Dependent variable.**   The dependent variable was perception of physicians towards electronic medical prescription system.

**Independent variables.**   The independent variables were sociodemographic factors (age, sex, profession, educational status, experience), system-related factors (perceived usefulness, perceived ease of use), organizational factors (infrastructure), and behavioural factors (knowledge, technical skill, previous IT experience).

**Operational definitions.**   In this study, "physician" includes general practitioners, residents, dentists, specialists, and subspecialists.

**Data processing and analysis.**   Data was entered using Epi-info version 7 and analysed using Statistical Package for Social Science (SPSS) version 20. Descriptive analyses were computed for the dependent variable in the study to determine the Perceptions of Physician to implement electronic e-Prescription. Adjusted odds ratios were used to measure the association of dependent and independent variables, 95% confidence intervals, and P-value was calculated to evaluate statistical significance. A value of P < .05, corresponding to a 95% CI, was considered statistically significant. Binary logistic regression was done and variables with p-value < = 0.2 were taken and multivariable logistic regression analyses were carried out to assess the effect of selected variables on perception to implement e-prescription. Standardized coefficient and 95% confidence intervals were calculated for each of the independent variables

in simple binary regression models with the perception of physicians to use electronic prescription as the dependent variable.

## Ethical consideration

In conducting the study, ethical clearance was obtained from the Debre Markos university ethical review board with the ethical approval number of HSC/R/C/Ser/Co/227/12/13. Additional permissions to access participants were also obtained from each hospital administrator. In addition, written informed consent was gained from all participants, participation in the study was voluntary, and no incentive was provided for the participants.

# Result

## Socio-demographic characteristics of participants

A total of 302 response was received (Out of 384 distributed questionnaire 302 valid response was received) with a response rate of 78.6%. About one-third 225(74.5%) of the respondents being male. The mean age was 28+_3.6SD years with the majority of the age group were 25–34. About 231(76.5%) of the physicians had a positive perception towards electronic prescription system Table 1.

Around 70.8% had computer usage experience for more than 5 years, 23.7% for 1–5 years and, the rest less than one year. As shown in Table 2, 97.3% were comfortable with the use of computers, and only 52% had used computers at the hospital. Of the 70.5% (181) who have heard of e-Prescriptions only 14% (37) had hands-on exposure in generating it Table 2.

About 90% of the participant claimed that their prescriptions to be legible and 76% liked paper prescriptions, on the other hand, 89% perceive that paper-based prescription is prone to error Fig 1.

The mean perceived usefulness and technical skill of physicians were 4±0.4, 3.9±0.45, respectively. As shown in Table 3 below, the majority of the participants perceive that using the e-Prescription system will decrease the cost of healthcare (71.2%) and 91% of the study participants think e-prescription would promote the use of data for research.

**Table 1. Sociodemographic characteristics of participants (n = 302).**

| VARIABLE | CATEGORY | FREQUENCY (#) | PERCENTAGE (%) |
|---|---|---|---|
| GENDER | Male | 225 | 74.5 |
| | Female | 77 | 25.5 |
| AGE | < = 30 | 165 | 54.7 |
| | >30 | 137 | 45.3 |
| EDUCATIONAL STATUS | General practitioner | 240 | 79.5 |
| | Resident | 39 | 12.9 |
| | Specialist | 23 | 7.6 |
| DEPARTMENT | Internal medicine | 105 | 34.7 |
| | Pediatrics | 54 | 17.9 |
| | Surgery | 59 | 19.5 |
| | Gynaecologist | 56 | 18.5 |
| | Dermatology | 14 | 4.6 |
| | Other | 14 | 4.6 |
| WORK EXPERIENCE | 1–3 Years | 192 | 63.6 |
| | 3–6 Years | 87 | 28.8 |
| | >6 Years | 23 | 7.6 |

**Table 2. Response of physician on current computer usage status (N = 302).**

| Question | Responses (%) | |
|---|---|---|
| | **Yes** | **No** |
| I am comfortable with computer use | 293 (97.3%) | 9(2.7) |
| I use computer for personal purpose | 297 (98.7%) | 5(1.3) |
| I use computer at home | 276 (91.3%) | 26(8.7) |
| I use computer at hospital | 155 (51.3%) | 147(48.7) |
| I have good knowledge of computer usage | 295 (97.7%) | 7(2.3) |

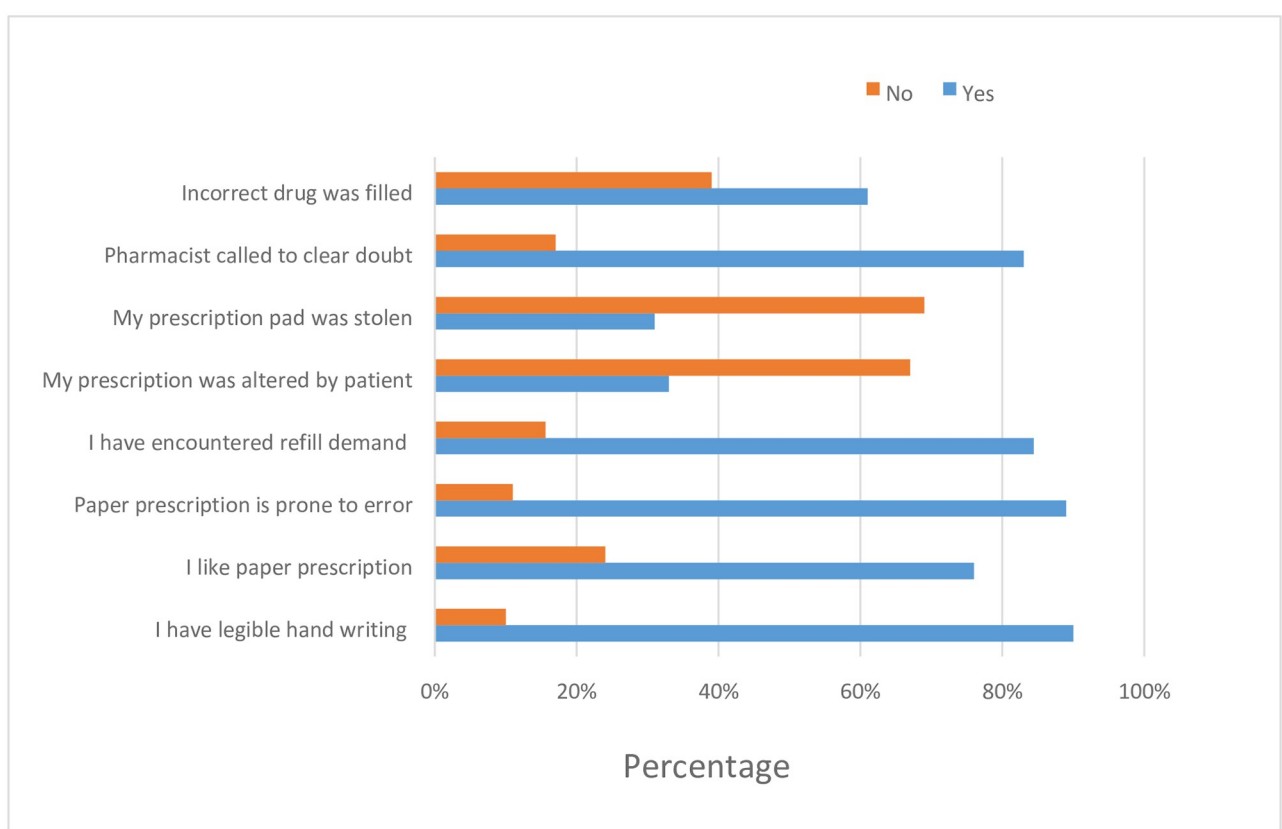

**Fig 1. Current prescription profile of participants (N = 301).**

**Table 3. Perception of physicians towards perceived usefulness of electronic prescription system (N = 302).**

| Variable | Disagree (1+2) | Neutral (3) | Agree (4+5) |
|---|---|---|---|
| Using an electronic prescription system will decrease the cost of healthcare | 7.9% | 20.9% | 71.2 |
| Using an electronic prescription system promote the use of data for research | 2% | 7% | 91% |
| The electronic prescription system will provide an alert when the patient receives medication | 3.3% | 16.7% | 80% |
| The electronic prescription system will improve alert about the drug | 0 | 6.3% | 93.7 |
| The electronic prescription system will save time and reduce error | 3% | 18.2% | 78.8% |
| The electronic prescription system will be safe | 3.3% | 18.2% | 78.5% |

Table 4. Perceived readiness of organization to implement e-prescription (N = 302).

| Variable | Disagree(1+2) | Neutral(3) | Agree(4+5) |
|---|---|---|---|
| I think the hospital firm can afford to improve the internet network | 19.5% | 29.5% | 51% |
| I think the internet network in the hospital is good | 45.4% | 16.2% | 38.4% |
| I think the hospital firm can afford to get all physicians computer | 40.1% | 36.1% | 23.8% |
| The hospital firm Can manage and be consistent with the system over a long time | 22.2% | 28.1% | 49.7% |
| I think the hospital firm would be ready to adopt the new system based on the infrastructural facilities | 25% | 45% | 30% |
| I think the patient will be ready to adopt the new system | 15.8% | 47.4% | 36.8% |

Table 4 depicts the perceived readiness of the institution to implement e-prescription. Even though the majority of the participants agreed to accept e-prescription once it was adopted in the institution, 40% felt that the hospital firm cannot afford to get all physician's computers. There was a compelling demand for infrastructure development and massive training in the Institution. Only 36.8% of the participants felt that the patient will be ready to adopt the e-prescription system.

## Factors associated with perceptions physicians towards electronic prescription

A total of 5 variables were selected as potential predictors for perception after bivariable logistic regression and entered into multivariable logistic regression. Variables included in multivariable logistic regression were the technical skill of physicians, internet access in the organization, gender, work experience and, perceived usefulness of the e-Prescription. The multivariable analysis of logistic regression pointed out the technical skill of the physicians, availability of internet access, perceived usefulness of the e-Prescription, and organizational factors as significant variables which were positively related to perception of physicians towards e-Prescription Table 5.

Table 5. Bivariable and multivariable logistic regression analysis of factors with perception among physicians at public health hospitals in Amhara region Northwest Ethiopia 2021(N = 302).

| Variable | Perception, n (%) | | Crude OR[a] (95% CI) | P value | AOR[b] (95% CI) | P value |
|---|---|---|---|---|---|---|
| | Positive | Negative | | | | |
| Current computer usage | | | | | | |
| Yes | 269 (89) | 19 (.062) | 3.86 (1–15.02) | .052 | 2.06 (0.51–8.31) | |
| No | 11 (0.04) | 3 (.01) | 1 | | 1 | |
| Internet access | | | | | | |
| Yes | 163 (54) | 2 (.66) | 13.846 (3.174–2.25) | .000 | 2.82 (1.75–4.50) | < .001 |
| No | 117 (38.7) | 20 (6.6) | 1 | | 1 | |
| Technical skill | | | | | | |
| Yes | 268 (88.7) | 12 (3.9) | 8.37 (2.8–25.21) | | 4.7 (1.27–17.41) | .002 |
| No | 16(5.3) | 6 (2) | 1 | | 1 | |
| Perceived usefulness | | | | | | |
| Yes | 265 (87.7) | 16(5.3) | 6.62 (2.3–19.3) | | 3.31 (1.01–12.12) | .041 |
| No | 15 (5) | 6 (2) | 1 | | 1 | |
| Organizational factor | | | | | | |
| Yes | 157(52) | 7(0.23) | 2.75(1.09–6.9) | .032 | 1.8(.66–5.07) | |
| No | 122(40.4) | 15(5) | 1 | 1 | | |

[a]OR: odds ratio.

[b]AOR: adjusted odds ratio.

## Discussion

This study was conducted to assess physicians' perceptions towards e-Prescription systems in the context of resource-limited hospitals. Insights were provided into the future successful implementation of the e-Prescription system since the perception of the user greatly affects it. The study was done among the prescribing doctors at public health hospitals, which have a high workload with a physician to population ratio of.0769/1000 [28]. The study revealed that almost all 89% perceive that paper-based prescription is prone to error and 84% of the respondents encountered refill demand due to the absence of data verification and validation in paper-based prescription.

The mean perceived usefulness and technical skill of physicians were 4±0.4, 3.9±0.45, respectively this result is slightly higher than the result 3.5 and 3.4 obtained in the prior study conducted in India [20]. This could be attributable to the time difference between the current study and the previous study, which was conducted five years earlier (2016). Physicians' perceptions about e-prescription were found to be influenced by factors such as internet availability, perceived usefulness, technical skill, and previous computer-related training. In this study, more than two-third 76.5% (95% CI 71.5–81) of the study participants had a positive perception towards e-Prescription. This result is consistent with that of a study done in Jordan (74.7%) [27].

Different studies conducted on technology acceptance in different domains have suggested that Perceived usefulness is the main determinant factor for new technology acceptance and utilization [29–31]. The finding of the current study was consistent with these studies. The current study revealed that the perceived usefulness of the e-prescription system is positively associated with the positive perception of physicians 3.31(95% CI 1.01–12.12). The odds of physicians who thought the e-prescription system as useful to have positive perception is 3.31 times more likely than their counterparts. The possible reason for this could be that, as the user thought the system would be useful more likely to have a positive perception.

According to this study, physicians who were working in an institution with internet access (WiFi) were 2.82 times more likely to develop positive perceptions than those who had no internet access (AOR 2.82, 95% CI 1.75–4.50).

The majority of authors who conducted a study on electronic information systems acknowledge that users' perception is more likely to be affected by their computer skills [17, 20, 31, 32]. The result of this study is consistent with those findings, this study revealed that physicians who had good technical skills were 4.7 times more likely to have a positive perception towards implementation of e-prescription than those who had poor technical skills (AOR 4.7, 95% CI 1.2717.41]).

This study summarizes the perceptions of front-line prescribers (physicians) as a pre-implementation assessment. Implementation of new information system greatly affected by Perception of front-line system users, as pre-implementation study the result suggests that implementation of e-Prescription would be successful within physicians' perception perspective. Even though the study population corresponds to a usual proportion of prescribers in the institution a larger sample size would have yielded better results, further more if included perception of pharmacists and patients would have provided better insights.

## Conclusion

The findings of this study showed that the positive perception of physicians towards e-Prescription was 76.5%. Good internet access, perceived usefulness of the e-prescription system, and the technical skill of the physicians were the most notable factors that were associated with a positive perception.

According to this study finding, the perception of the user has a significant impact on the successful implementation of an e-prescription system. To be successful in e-prescription system implementation, health care organizations should improve the computer technical skill of physicians' providing training on digital device usage, and improve internet connectivity. As a limitation, the study was a cross-sectional which has inherited limitation of a cross-section study. In addition to this the sample size of the study was a little bit smaller. Furthermore, there might also be a possibility of bias towards young physicians. In addition to this the sample size of the study was a little bit smaller.

## Supporting information

**S1 File.**
(DOCX)

**S1 Data.**
(SAV)

## Acknowledgments

We would like to thank Debre Markos University institute of public health for the approval of the ethical clearance, Respective hospitals and directors for giving us permission to collect data. We also forward gratitude to data collectors, supervisors and study participants.

## Author Contributions

**Conceptualization:** Gizaw Hailiye Teferi, Zegeye Regasa Hordofa.

**Data curation:** Gizaw Hailiye Teferi, Tewodros Eshete Wonde, Maru Meseret Tadele, Bayou Tilahun Assaye, Zegeye Regasa Hordofa.

**Formal analysis:** Gizaw Hailiye Teferi, Tewodros Eshete Wonde, Maru Meseret Tadele, Mohammedjud Hassen Ahmed, Samuel Hailegebrael.

**Funding acquisition:** Gizaw Hailiye Teferi.

**Investigation:** Gizaw Hailiye Teferi.

**Methodology:** Gizaw Hailiye Teferi, Tewodros Eshete Wonde, Maru Meseret Tadele, Bayou Tilahun Assaye, Zegeye Regasa Hordofa, Samuel Hailegebrael.

**Project administration:** Gizaw Hailiye Teferi.

**Resources:** Gizaw Hailiye Teferi.

**Software:** Gizaw Hailiye Teferi.

**Supervision:** Gizaw Hailiye Teferi, Bayou Tilahun Assaye, Mohammedjud Hassen Ahmed.

**Validation:** Gizaw Hailiye Teferi, Maru Meseret Tadele, Mohammedjud Hassen Ahmed.

**Visualization:** Gizaw Hailiye Teferi, Bayou Tilahun Assaye.

**Writing – original draft:** Gizaw Hailiye Teferi, Tewodros Eshete Wonde, Maru Meseret Tadele, Bayou Tilahun Assaye.

**Writing – review & editing:** Gizaw Hailiye Teferi, Maru Meseret Tadele.

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
