## [Decision Letter · Decision Letter 0]

5 Aug 2021

PONE-D-21-14066

Perception of physicians towards electronic prescription system and associated factors at resource limited setting 2021: Cross sectional study

PLOS ONE

Dear Dr. Hailiye,

Thank you for submitting your manuscript to PLOS ONE. After careful consideration, we feel that it has merit but does not fully meet PLOS ONE’s publication criteria as it currently stands. Therefore, we invite you to submit a revised version of the manuscript that addresses the points raised during the review process.

We look forward to receiving your revised manuscript.

Kind regards,

Dylan A Mordaunt

Academic Editor

PLOS ONE

A clean copy of the edited manuscript (uploaded as the new *manuscript* file).

4. Please amend the manuscript submission data (via Edit Submission) to include author Maru Meseret, Bayou Tilahun , Zegeye Regasa , Tewodros Eshetie , Mohammedjud Hassen , Samuel Hailegebriel

5. Thank you for submitting the above manuscript to PLOS ONE. During our internal evaluation of the manuscript, we found significant text overlap between your submission and the following previously published works, some of which you are an author.

- https://mhealth.jmir.org/2021/3/e19310 (Abstract)

- https://www.dovepress.com/surgical-and-medical-error-claims-in-ethiopia-trends-observed-from-125-peer-reviewed-fulltext-article-MB (Intro)

- https://www.jyoungpharm.org/sites/default/files/JYoungPharm_10_3_313.pdf (results)

Please revise the manuscript to rephrase the duplicated text, cite your sources, and provide details as to how the current manuscript advances on previous work. Please note that further consideration is dependent on the submission of a manuscript that addresses these concerns about the overlap in text with published work.

Additional Editor Comments (if provided):

Please see the feedback from these editors. My apologies, I unintentionally invited a number of the backups as initial reviews which is why there are so many, though I think the feedback will be useful for amending your submission and preparing responses.

Reviewers' comments:

Reviewer's Responses to Questions

**Comments to the Author**

1. Is the manuscript technically sound, and do the data support the conclusions?

Reviewer #1: No

Reviewer #2: Yes

Reviewer #3: Partly

Reviewer #4: Yes

Reviewer #5: Yes

Reviewer #6: Yes

Reviewer #7: Yes

Reviewer #8: Yes

Reviewer #9: Partly

Reviewer #10: Yes

Reviewer #11: Yes

Reviewer #12: No

Reviewer #13: No

Reviewer #14: No

2. Has the statistical analysis been performed appropriately and rigorously? 

Reviewer #1: Yes

Reviewer #2: Yes

Reviewer #3: No

Reviewer #4: I Don't Know

Reviewer #5: I Don't Know

Reviewer #6: Yes

Reviewer #7: Yes

Reviewer #8: Yes

Reviewer #9: Yes

Reviewer #10: I Don't Know

Reviewer #11: No

Reviewer #12: No

Reviewer #13: No

Reviewer #14: No

3. Have the authors made all data underlying the findings in their manuscript fully available?

Reviewer #1: Yes

Reviewer #2: No

Reviewer #3: Yes

Reviewer #4: Yes

Reviewer #5: No

Reviewer #6: Yes

Reviewer #7: Yes

Reviewer #8: Yes

Reviewer #9: Yes

Reviewer #10: Yes

Reviewer #11: Yes

Reviewer #12: Yes

Reviewer #13: No

Reviewer #14: No

4. Is the manuscript presented in an intelligible fashion and written in standard English?

Reviewer #1: No

Reviewer #2: Yes

Reviewer #3: Yes

Reviewer #4: Yes

Reviewer #5: No

Reviewer #6: Yes

Reviewer #7: Yes

Reviewer #8: Yes

Reviewer #9: No

Reviewer #10: No

Reviewer #11: No

Reviewer #12: No

Reviewer #13: No

Reviewer #14: No

5. Review Comments to the Author

Reviewer #1: This paper talks about the Perception of physicians towards electronic prescription system and associated factors at resource limited setting 2021 using cross sectional study design. The idea of this manuscript was okay but the execution of the study wasn’t. The abstract is vague and condensed with extra information such as the method part. There is a lack of support from previous studies in introduction and discussion. It wasn’t clear how would the authors did the reliability test on such a yes/no type of questions! It seems they do the study in a hurry. The type of questions were driving the answer toward “yes” answer and seems there is a bias. Enormous amount of English mistakes grammatically and misspelling.

Reviewer #2: The adoption of e-prescription platforms is increasing worldwide and is already considered standard and mandatory in many countries. The usefulness and benefits of electronic prescriptions are well described, but the viability of its implementation, notably in less resourceful countries, still needs to be evaluated.

The authors conducted a well-designed questionnaire-based inquiry in an underserved area to evaluate the local physicians' perception of computer use and preparedness for e-prescription platforms implementation. However, there are some aspects of the manuscript that need critical, major, and minor revisions, concerning mainly scientific standards and methods description:

Critical (authors must provide data/modifications):

1) There is no mention of ethical committee approval and informed consent provided by subjects.

2) Data is not fully available (please upload to approved platforms and provide the links).

3) Methods should describe how the investigators sent/filled the questionnaires (web-based, paper, or other)

Major revision: (modifications needed for publication)

1) The questionnaire should be annexed as applied in its full version (translated from Amharic if needed).

2) With the complete questionnaire annexed, the description in the "methods" section can be rewritten with a more general description.

3) There are many types of e-prescription platforms. The authors need to describe what type of e-prescription is being asked about, such as:

- printed form with physical signature,

- printed form with verifiable digital code,

- electronic form sent directly to the hospital network pharmacy or central prescription portal,

- full electronic version (PDF) available for patients/physicians in web portals or send by e-mail,

- weblink or code sent by SMS/text platforms,

- need for verified digital signatures or public key certificates

Minor revisions:

- Grammar and writing style review

- Tables style review

Reviewer #3: The English language quality is generally good, but a small number of corrections are still needed.

The sample size computation is written in a somewhat confused manner and the software used for computation is not stated (it should be). Besides, a 10% non-response rate is overly optimistic. Moreover, it is my understanding that although the sample size was computed using a simple random sampling, whereas the sampling process was stratified. Conventionally scientists use 80% power (to detect a difference), rather an 80% sample. We have also doubts about the lottery system, because that would imply that the researcher already knows the name of all potential participants; if so, the authors should first explain how did they obtain the names (or other identification means) of all participants, so as to select 384.

The description of the questionnaire could probably be better replaced with a full copy of it annexed to the paper, and only a summary being described in the main text.

Apparently the study is underpowered, because the authors only recruited 302 respondents (despite the computed sample size of 384). Moreover, as mentioned above, the non-response rate was over 21% and not 10%.

“The mean age was 28+_3.6SD years with the majority of the age group were 25-34”. We tend to be skeptical of the representativity of the sample and of the correctness of the sampling process, because there seems to be a strong bias towards young doctors (as confirmed by the work experience of less than 4 years for almost 64% of the sample). It is very unlikely that the hospitals involved are working with 64% of their staff having less than 4 years of experience and only less than 8% having more than 6 years of experience. The authors should acknowledge this bias as a strong limitation, and avoid extrapolation to other settings or ages, for which the sample has little representativity.

Table 4: adjusted odds ratios values are reported, but there is no information on the models developed (only one? Several?) or the variables included in (each of) the models. Besides, p values should be reported even in the case they are larger than 0.05, and 0.000 should be replaced with p<0.001.

We are afraid that the interpretation of the 4.7 OR (“4.7 times more likely to have a positive perception…”) is erroneous and should be corrected (see this paper for an explanation of why: https://www.ncbi.nlm.nih.gov/pmc/articles/PMC5253299/)

Reviewer #4: Minor comments:

1. Please review manuscript carefully for grammatical errors and typos such as:

a. In abstract, 4th sentence should start as this study (and not Thus is study...)

b. In abstract, results first sentence: 76.5% should be outside bracket

c. Under methods section (Study design and settings), second sentence describing sample size, is 384 80% of total physician population? Current sentence indicates that 384 is 80% of total population in the region. Please correct.

2. Define medical error in introduction and include the most common medical errors observed in your region.

3. Even though perceived usefulness of electronic prescription was assessed, the questionnaire does not include a question regarding prior awareness of the physician about electronic prescription, how it works and its benefits. This is crucial question that needs to be used to understand any biases associated with their results.

4. the manuscript does not discuss potential pitfalls/shortcomings of the study. Please include this information.

Reviewer #5: PONE-D-21-14066

Perception of physicians towards electronic prescription system and associated factors at resource limited setting 2021: Cross sectional study

Reviewer’s Comment

General Comments:

Throughout the manuscript, there are numerous grammatical faults. It is strongly recommended that the work be edited by a native English speaker or a company that provides editing services.

Key words: The authors must ensure that the keywords are MeSH words and that they are in alphabetical order.

Introduction:

According to the findings of this study, 89 percent of participants believe that paper-based prescriptions are prone to errors, however the literature mentioned [6] reveals that paper-based prescriptions have been a favored communication method for many years for physicians.

Isn't it true that if a paper-based prescription is prone to drug errors, it shouldn't be preferred?

Several references have been provided discussing paper-based versus electronic prescriptions, but despite several studies conducted on the topic in various regions of the world, there is not a single source about physicians' perceptions of electronic prescriptions.

Methods:

Incorporating a formula for calculating sample size would be more useful.

A total of 17 referral hospitals were chosen, with each receiving a proportional allotment. What was the foundation for the proportional allocation and how was it done?

“Do you have legible handwriting?” was asked as a yes or no question to assess current prescription practice. I believe that observing written prescriptions rather than asking questions is a better way to evaluate this assertion.

Technical skill was assessed using questions like: do you think using a computer for an

electronic prescription would make your work more effective and accurate?, are you good at

operating a computer system?, Are you fast in responding to training on a new device? Are you

really motivated to pick up the new electronic prescription system (1. Strongly disagree, 2.

Disagree 3. Neutral 4. Agree 5. . Strongly agree). How might these types of questions be used to measure technical skills? For determining attitude or perception, likert scale questions are utilized.

More information on the expert panel (10 doctors), such as their qualifications, departments, and so on, should be included.

What necessary correction were done in questionnaire based on the pre-test finding?

Who gave two-days training for five data collectors) on the objective of the study and data

collection procedures? Who were the data collectors? How were their levels of education, experience, and other factors matched to ensure that the data they collected was uniform?

Please specify the number and date of the ethical approval.

Results:

Table 5 might be presented in a more appealing manner.

Discussion:

In light of the outcomes of this study, this area demands greater critique and clarity.

Conclusions:

Availability of data and materials: It is recommended that they be made available in the journal's repository.

References:

Check the journal's requirements for references and their citations in the article.

Reviewer #6: Comments to authors

1- This sentence in the abstract need to be revised Grammarly “ This medical error is even more serious in developing countries that lack a technologically supported healthcare system”

2- Please rewrite this sentence and organize it “ In Ethiopia because of the traditional paper-based

prescription system the overall medication prescribing error rate of 58.07%, in which incomplete

prescriptions and dosing errors were the most prevalent error types which could be prevented if there were an electronic prescription system”

3- I gave 2 examples of grammars errors I suggest to check all manuscript language to make the language more academic. There are several grammars errors the manuscript need to be checked Grammarly.

4- “ A study released in 2016 found that medical error is the third leading cause of death in the United States, after heart disease and cancer.” Please add reference after this sentence

5- “ the medical death rate” this phrase is not accurate I suggest called it medical errors related death rate

6- “ Medication errors are the leading causes of avoidable patient harm in the health care system across the world” add reference

7- “ the overall medication error in Ethiopia was found to be 57.6%” this sentence is not clear 57.6% from total prescriptions? Or from total medical errors or ….

8- “ Indeed, studies show significant improvements associated with e-prescription implementation, including an 86% decrease in serious medication errors, and an increase in Medicare formulary adherence from 14% to 88% [8].” Authors started with studies show… but only one reference was added

9- In study design and settings section, there are many unneeded details that can be shown in the questioner directly, I suggest to reduce the details about the survey and just refer the reader to the questionnaire in the supplementary files. Just summarize the domains of the survey

10- “ ethical clearance was obtained from the Debre Markos university ethical review board.” I suggest adding the number of ethical approval letter

11- In table 5, the p value of (Current computer usage) is not written

12- “ The study was done amongst the prescribing doctors at public health hospitals which has a high workload with physicians to population ratio of .0769/1000” I suggest change amongst to among. Additionally, I don’t know the meaning of numbers followed the sentence.

13- “ The finding of the current study was consistent with this studies” this sentence in discussion has grammar mistakes

Reviewer #7: This study aims to assess physician’s perception towards electronic prescription system implementation in Ethiopia. This study administered questionnaires to 384 physicians from several hospitals in certain regions in Ethiopia to gauge their perception of electronic prescription systems.

This work covers an interesting topic. Below I have a few minor comments

Minor comments

1. In the Introduction section, it is stated that “A study released in 2016 found that medical error is the third leading cause of death in the United States, after heart disease and cancer”. However, no citation is given.

2. In the future, I think it will be interesting to include some space in these questionnaires where physicians and others responding to the questionnaires can write (in their own words) some of their perception to electronic prescription systems. With this, the researchers conducting the study can check to see if there are any common themes and if these themes vary or are similar to some of the questions being asked in the questionnaire

Reviewer #8: The manuscript is well written with sound idea and presentation. All parts of manuscript are mentioned. Methodology is well explained with sample size and states applied. The questionnaire is not given separately, it must be given in the paper. Discussion is not enough for this paper. Discuss it with more references.

Reviewer #9: I thank the authors for the interesting manuscript and my comments are listed below

the title is clear with no edits needed

the abstract is has some grammar mistakes

the introduction has many comparative studies that needs to be moved to the discussion

suggest to be consistent when using decimals in %, in intro remove decimals

the intro is too long with un-necessary details

References need to be combined using reference management tool e.g. (10), (11), (12) to (10-12)

Methods

lots of grammar mistakes

sampling is confusing need to be clarified

what do you mean by reliability was it calculated after the pilot, you said that the expert physicians only reviewed it

was the survey paper base?

was the survey in English?

Results

total sample 302, was not mentioned in the abstract

why was the cut-off point in age was 30 years?

tables need more organization

I don't understand why the Likert scale in tables were combined? 1+2 and 4+5

the regression table could be presented in a different way, the current way is confusing

Reviewer #10: I think this manuscript has a merit but there are many comments that must be addressed. ‎

I have found some errors that should not be present at this level, for example there are 2 ‎references in bibliography but were missed in the next. ‎

The authors must read again the Author’ guidelines and fill all the requirements. ‎

The authors must proofread their work by an English language professional.‎

My specific comments are included in the PDF file.‎

Reviewer #11: The author carried out a cross sectional study on Perception of physicians towards electronic prescription system and associated factors at resource limited setting 2021. However the author needs to resolve the queries raised during review it.

1. A lot of grammatical errors present through out the manuscript. Author needs to check it by a native English speaker.

2. Why does the author select the physicians with age below 35? Need to clarify it.

3. Why the female participants were in a ratio of 25% of total participants (302).

4. What were the strength and limitations of this study must incorporate in conclusion section.

5. Author should include reference no. of ethical permission.

Reviewer #12: The submitted manuscript requires following rectifications in addition to language repeat:

1) The title may be revised "Instead of resource limited setting some proper wording may be placed.

2) In introduction first define the medical error, and then switch to the medication error.

3) Authors should also explain properly statistical out put.

4) While putting the p-value use zero before decimal and values after the decimal should be synchronous.

5) Place all the abbreviations in foot notes of the tables.

Reviewer #13: 1.0. The data presented may support the conclusion but there is much to desire and replicating this study would be a challenge

2.0 I have challenges with how the analyses were done and results were presented.

3.0 Did not see how some means were calculated

4.0. The presentation of some aspect especially the methodology found a huge cap there

Reviewer #14: The study topic itself is interesting from a health systems research perspective. However, several factors limit the enthusiasm for this manuscript. Firstly, the survey questions seem to elicit cross-sectional responses regarding uptake of and perceptions about e-prescribing and EHR use. The questions themselves do not appear to have been sufficiently tested. For e.g. ~90% respondents answered Yes to the statement "Paper prescription is prone to error" and about 70% responded that they like paper prescriptions while 79% responded that electronic prescriptions would reduce errors and save times. All three of these statements contradict each other. It is unclear to this reviewer what the general attitudes of this survey sample are and how accurately do they reflect upon behaviors. There is little mention of current trends in e-prescribing and the availability of systems at the various study sites. This needs to be clearly explained. Organizational factors such as administrative assistance technical help etc. are not addressed. The logistic regression seems inappropriate since the outcome variable is not measurable (perception). It is unclear how perception, the outcome variable was scored as 0/1/for the logistic regression on a 5-point Likert scale. The authors seem of to overreach in their analyses by stating that internet access had a positive perception. This statement doesn't mean much if the access is inconsistent. However, it is unclear since internet access and systemic factors are not well-described. The study has significant gaps in methodology and survey design and analysis.

6. PLOS authors have the option to publish the peer review history of their article (what does this mean?). If published, this will include your full peer review and any attached files.

Reviewer #1: No

Reviewer #2: No

Reviewer #3: No

Reviewer #4: **Yes: **sandeep artham

Reviewer #5: **Yes: **Mukhtar Ansari

Reviewer #6: **Yes: **Abdullah Salah Alanazi

Reviewer #7: No

Reviewer #8: No

Reviewer #9: No

Reviewer #10: **Yes: **Salem Abukres

Reviewer #11: **Yes: **AHM Khurshid Alam

Reviewer #12: No

Reviewer #13: No

Reviewer #14: No

---

## [Author Response · Author response to Decision Letter 0]

2 Oct 2021

Here is a point-by-point response to the reviewers’ comments and concerns. 

 Comments from reviewer 1

Comment 1: It wasn’t clear how the authors would did the reliability test on such a yes/no type of questions!

Response: Thank for pointing out such comments, but most of the question in the study were Likert scale type, beside this reliability/validity test can be done for Yes or No type questions.

 Comments from Reviewer 2

 Critical comments by reviewer 2

 Comment 1: There is no mention of ethical committee approval and informed consent provided by subjects.

 Response: Thanks for the comment but we have already included this point under Ethical consideration section of the manuscript at page 8. 

 Comment 2: Data is not fully available (please upload to approved platforms and provide the links)

 Response: We will provide all the data needed

 Comment: Methods should describe how the investigators sent/filled the questionnaires (web-based, paper, or other)

 Response: As it was already stated in method section page 7 the data was collected using self-administered questionnaire(data collector provided paper printed questionnaire for participants)

Major comments by reviewer 2

 Comment 1: The questionnaire should be annexed as applied in its full version (translated from Amharic if needed)

 Response: we have provided the questionnaire under annex section page but we the questionnaire wasn’t translated to in Amharic we used the English questionnaire as it is. 

 Comment 2: With the complete questionnaire annexed, the description in the "methods" section can be rewritten with a more general description.

Response: thanks for reminding us to revise this section we made the revision and highlighted the change. 

 Comment 3: There are many types of e-prescription platforms. The authors need to describe what type of e-prescription is being asked about

Response: Thank you for raising such important question, the type e-Prescription method asked about is an electronic form sent directly to the hospital network pharmacy. 

Comments from Reviewer 3

Comment 1: Sample size calculation is not clear

Response: Thanks for the question again, to clarify the issue we used single population proportion formula to calculate the sample size. We took 50% because there was no previous study conducted in the study area.

Sample size (n) =((〖Z α⁄2)〗^2 x p(1-P))/d^2 , (n) =((〖1.96)〗^2 x 0.5(1-0.5))/〖(0.05)〗^2 = 384.2

Where;

 n=estimated sample size

 p=single population proportion (50%).

 Zα/2 =is value of standard normal distribution (Z-statistic) at the 95% confidence level (α=0.05) which is 1.96,

 d=is the margin of error 5% (0.05) 

Comment 2: The authors should first explain how they obtained the names of all participants, so as to select 384.

Response: After the sample size was determined proportional allocation was made for each hospital and the names of the physicians was collected from HRIS office simple random sampling was made

Comment 3………authors only recruited 302 respondents (despite the computed sample size of 384). Moreover, as mentioned above, the non-response rate was over 21%andnot10%.

Response: Your point here is valid and acceptable, we agree that the statement in the result section which state” A total of 302 physicians were included” is a little bit confusing.

Here we mean that a total of 302 valid response were received from participant (Total of 367 response were received but 52 of the response were incomplete and the other 13 contains some personal identification like name of the physician so we discarded 65 responses), in addition to this there were 17 questionnaires which were not returned back. But we distributed 384 questionnaires

Comment 4…there seems to be a strong bias towards young doctors, the authors should acknowledge this bias as a strong limitation

Response: This is an important issue and we accept your point to some extent and we acknowledge the bias under limitation section in the revised manuscript. But there was undeniable reason behind this, as stated in the previous manuscript the study was done in two zones namely East Gojjam and west Gojjam which has 17 public hospitals (1 referral hospital, 2 General hospitals and 14 primary hospitals). So most of the study participants were from primary hospitals, and the minimum physician professional requirement of primary hospital in Ethiopia is general practitioner. That was the case.

Comment 5: adjusted odds ratios values are reported, but there is no information on the models developed (only one? Several?)..

Response: Thank for pointing out such critical point, we have mentioned that we done binary logistic regression and multivariable logistic regression but it lacks detail so we have included the detail under data processing and analysis section line 6 to 10. 

Comments from Reviewer 4

Comment 1: Define medical error in introduction and include the most common medical errors observed in your region.

Response: Thank for suggesting this important point, we incorporated your suggestion in the introduction section line 5-7. 

Comment 2: Even though perceived usefulness of electronic prescription was assessed, the questionnaire……

Response: Your point is valid but we have already collected the data, at this time we can’t include your suggestion. Beside this we have used perceived usefulness questionnaire from Technology acceptance model which doesn’t include previous experience questions.

Comments from Reviewer 5

Comment 1: According to the findings of this study, 89 percent of participants believe that paper-based prescriptions are prone to errors, however the literature mentioned [6] …………. Isn't it true that if a paper-based prescription is prone to drug errors, it shouldn't be preferred

Response: we agree that this needs clarification, the point here is before advancement of information communication technology physicians used to prescribe drug using paper printed forms which is prone to different errors. The article 6 in the manuscript shows the history drug prescription before the invention of e-Prescription. The article did not compare the paper based prescription method with e-prescription.

Comment 2: Incorporating a formula for calculating sample size would be more useful.

Response: Thank you this valuable suggestion we have incorporated sample size calculation formula in the revised manuscript in the method section line 8-16.

Comment 3: A total of 17 referral hospitals were chosen, with each receiving a proportional allotment. What was the foundation for the proportional allocation and how was it done?

Response: As you already mentioned there are 17 hospitals (1 referral hospital, 2 general hospitals and 14 primary hospitals). In order ensure the representativeness of the sample we used proportional allocation for each hospitals.

In order to show how proportional allocation done we provide an example below

One of these hospital is Debre Markos hospital which has 132 physicians, there are a total of 465 physicians and the sample size for this study is 384. So using proportional allocation 

(Total number of physcian in Debre markos hospital(132)*Total sample needed(384))/(Total number of physcians in 17 hospitals (465))= 109 participants were selected from Debre Markos referral hospital.

This formula was used to allocate sample size for all other hospitals.

Comment 4: “Do you have legible handwriting?” was asked as a yes or no question to assess current prescription practice. I believe that observing written prescriptions rather than asking questions is a better way to evaluate this assertion.

Response: Yes it would be more appropriate if we observe rather than asking them, but as stated in the manuscript we used self-administered questionnaire to collect the data. That’s why we couldn’t observe their hand writing.

Comment 5: More information on the expert panel (10 doctors), such as their qualifications, departments, and so on, should be included. 

Response : Thank you for this suggestion, based on your comment we have revised the document and your suggestion is included under the method section 

Comment 6: What necessary correction were done in questionnaire based on the pre-test finding?

Who gave two-days training for five data collectors) on the objective of the study and data

collection procedures? Who were the data collectors? How were their levels of education, experience, and other factors matched to ensure that the data they collected was uniform?

Response: corrections like grammatical errors, order of the questions, were corrected.

The training were provided principal investigator, data collectors were Bsc health informatics professionals with two years of working experience.

Comment 7: Please specify the number and date of the ethical approval.

Response: This is incorporated in the revised manuscript.

Comment 8: It is recommended that data and material made available in the journal's repository.

Response: We will upload the data with revised document.

 Comments from Reviewer 6

Comment 1: A study released in 2016 found that medical error is the third leading cause of death in the United States, after heart disease and cancer.”

Response: Thank you for reminding us we missed this reference, based on your comment the reference is added.

Comment 2: Medication errors are the leading causes of avoidable patient harm in the health care system across the world” add reference

Response: Thank you for reminding us we missed this reference, based on your comment the reference is added.

Comment 3: “ the overall medication error in Ethiopia was found to be 57.6%” this sentence is not clear 57.6% from total prescriptions? Or from total medical errors or ….

Response: 57.6% represents the total medical errors including drug administration and prescription. 

Comment 4: “Indeed, studies show significant improvements associated with e-prescription implementation, including an 86% decrease in serious medication errors, and an increase in Medicare formulary adherence from 14% to 88% [8].” Authors started with studies show… but only one reference was added

Response: Thanks for the suggestion. It was editorial error and it changed to study.

Comment 5: In study design and settings section, there are many unneeded details that can be shown in the questioner directly, I suggest to reduce the details about the survey and just refer the reader to the questionnaire in the supplementary files. Just summarize the domains of the survey

Response: Based on your suggestion the questions were removed in summarised under Appendix section.

Comment 7: In table 5, the p value of (Current computer usage) is not written

Response: Of course it is not written. We left it black because current computer prescription was not a significant variable based on multivariable logistic regression. Its p value was >.05.

 Comments from Reviewer 7

Comment 1: In the Introduction section, it is stated that “A study released in 2016 found that medical error is the third leading cause of death in the United States, after heart disease and cancer”. However, no citation is given. 

Response: Response: Thank you for reminding us we missed this reference, based on your comment the reference is added.

 Comments from Reviewer 9

Comment 1: what do you mean by reliability was it calculated after the pilot, you said that the expert physicians only reviewed it.

Response: Experts of physicians with different speciality reviewed the tool or the questionnaire before data collection. Beside that we have conducted pilot study outside of 17 hospitals, and reliability test were performed, it was found to be 0.78.

Comment 2: was the survey paper base? was the survey in English?

Response: Of course it was paper based and the questionnaire was in English language.

Comment 3: total sample 302, was not mentioned in the abstract

 Response: Your point here is valid and acceptable, we agree that the statement in the result section which state” A total of 302 physicians were included” is a little bit confusing.

Here we mean that a total of 302 valid response were received from participant (Total of 367 response were received but 52 of the response were incomplete and the other 13 contains some personal identification like name of the physician so we discarded 65 responses), in addition to this there were 17 questionnaires which were not returned back. But we distributed 384 questionnaires. But the total sample was 384.

Comments from reviewer 11

Comment 1: Why does the author select the physicians with age below 35? Need to clarify it

Response: Thanks for asking as for the clarification, as we have already stated in manuscript under result section the majority of the age group were 25-34, but it doesn’t meant that all the participants are under the age of 35. There were participants with age of above 35. The majority of the age group were 25-34 because most of the study participants were from primary hospitals, and the minimum physician professional requirement of primary hospital in Ethiopia is general practitioner which is fresh graduate that tend to be younger.

Comment 2: Why the female participants were in a ratio of 25% of total participants (302).

Response: As you have said there are only 25% female participants. This the output of the data collected.

Comment 3: What were the strength and limitations of this study must incorporate in conclusion section.

Response: Thank you for the suggestion. We have incorporated your suggestion under “limitation of the study” section.

 Comments from Reviewer 12

Comment 1: In introduction first define the medical error, and then switch to the medication error.

Response: Thank you for reminding us such an important point missed in the previous document, we incorporated the definition of Medical error under introduction section line 5 to 7.

Comment: While putting the p-value use zero before decimal and values after the decimal should be synchronous.

Response: That was an error and it corrected.

Comments from Reviewer 13

Comment 1: I have challenges with how the analyses were done and results were presented.

Response: Thank you for taking you time to review our paper. But your point here is too general, under method section we have stated how the analysis were performed, the software used and the model fitted. Please make sure that you raised specific that needs clarification so that we can address your comment. 

Comment 2: Did not see how some means were calculated

Response: We will look forward for your specific question on which mean calculation need clarification, otherwise it is difficult to address this question.

Comment 3: The presentation of some aspect especially the methodology found a huge cap there

Response: Thank you, we have made some revision on the methodology section. But if you still have additional issue on the revision we look forward to give clarification for your comments.

Response for general comment from reviewer 14

…….90% respondents answered Yes to the statement "Paper prescription is prone to error" and about 70% responded that they like paper prescriptions while 79% responded that electronic prescriptions would reduce errors and save times. All three of these statements contradict each other

Response: Thank you raising such an important comment, we agree that the statement may contradict but the question are different someone may like paper prescription because it is easy and He/She has used it many years. In addition to this they may have no skill and ability to use e-Prescription. Due to these reason they like traditional paper based prescription, but doesn’t mean that this paper based prescription is not prone to error. They like it but they acknowledge its limitation. 

The third question is under perceived usefulness of e-Prescription section that shouldn’t compared with the others. 

The logistic regression seems inappropriate since the outcome variable is not measurable (perception). It is unclear how perception, the outcome variable was scored as 0/1/for the logistic regression on a 5-point Likert scale.

Response: Even though the original questionnaire was 5-point Likert scale we have transformed the data in to two category which 0 and 1 negative and positive respectively. Based this result logistic regression was performed.

…… stating that internet access had a positive perception. This statement doesn't mean much if the access is inconsistent. However, it is unclear since internet access and systemic factors are not well-described

Response Here we compared perception of physicians working at hospitals those have internet connection and those has no internet connection. Based on this the result revealed that physicians in organization with internet connection to develop positive perception. We haven’t considered other factors.

---

## [Decision Letter · Decision Letter 1]

7 Dec 2021

PONE-D-21-14066R1

Perception of physicians towards electronic prescription system and associated factors at resource limited setting 2021: Cross sectional study

PLOS ONE

Dear Dr. Hailiye,

Thank you for submitting your manuscript to PLOS ONE. After careful consideration, we feel that it has merit but does not fully meet PLOS ONE’s publication criteria as it currently stands. Therefore, we invite you to submit a revised version of the manuscript that addresses the points raised during the review process.

We look forward to receiving your revised manuscript.

Kind regards,

Dylan Mordaunt

Academic Editor

PLOS ONE

Journal Requirements:

Reviewers' comments:

Reviewer's Responses to Questions

**Comments to the Author**

1. If the authors have adequately addressed your comments raised in a previous round of review and you feel that this manuscript is now acceptable for publication, you may indicate that here to bypass the “Comments to the Author” section, enter your conflict of interest statement in the “Confidential to Editor” section, and submit your "Accept" recommendation.

Reviewer #1: (No Response)

Reviewer #3: (No Response)

Reviewer #4: All comments have been addressed

Reviewer #5: (No Response)

Reviewer #6: All comments have been addressed

Reviewer #7: All comments have been addressed

Reviewer #8: (No Response)

Reviewer #11: All comments have been addressed

Reviewer #12: All comments have been addressed

2. Is the manuscript technically sound, and do the data support the conclusions?

Reviewer #1:  Partly

Reviewer #3: Partly

Reviewer #4: Yes

Reviewer #5: Yes

Reviewer #6: Yes

Reviewer #7: Yes

Reviewer #8: Yes

Reviewer #11: Yes

Reviewer #12: Yes

3. Has the statistical analysis been performed appropriately and rigorously? 

Reviewer #1: Yes

Reviewer #3: No

Reviewer #4: Yes

Reviewer #5: I Don't Know

Reviewer #6: Yes

Reviewer #7: Yes

Reviewer #8: Yes

Reviewer #11: Yes

Reviewer #12: Yes

4. Have the authors made all data underlying the findings in their manuscript fully available?

Reviewer #1: Yes

Reviewer #3: Yes

Reviewer #4: Yes

Reviewer #5: No

Reviewer #6: Yes

Reviewer #7: (No Response)

Reviewer #8: Yes

Reviewer #11: Yes

Reviewer #12: Yes

5. Is the manuscript presented in an intelligible fashion and written in standard English?

Reviewer #1: No

Reviewer #3: Yes

Reviewer #4: Yes

Reviewer #5: No

Reviewer #6: Yes

Reviewer #7: (No Response)

Reviewer #8: Yes

Reviewer #11: Yes

Reviewer #12: Yes

6. Review Comments to the Author

Reviewer #1: The manuscript needs extensive English editing. Some of the required questions and reviewer comments were not addressed appropriately.

Reviewer #3: Although the sample size calculation has been explained, it is obvious that the authors underecruted their subjects (they submitted the questionnaires to 384 potential respondents, but only 302 provided valid responses). They should have sent the questionnaires to 450-500 questionnaires in order to get a valid sample size of at least 384. As such, the inappropriate sample size should be acknolwedged as one of the limitations of this study.

Reviewer #4: Authors have thoroughly gone through all the reviewers comments and have addressed all comments successfully…..

Reviewer #5: -Keywords are still not in alphabetical order.

-I am still not convinced with accessing current prescription practice through the question "do you have legible handwriting?" In general, i believe that no one will say that his handwriting is illegible!

-Proportional allocation of physician/hospital need to elaborate in the manuscript as well.

-Ethical approval number and date need to be specified in the manuscript.

-Plenty of grammatical errors were still noticed throughout the manuscript.

Reviewer #6: (No Response)

Reviewer #7: (No Response)

Reviewer #8: The manuscript is describing the research precisely. This paper provides sound information about e-prescription and variables affecting the system. Methodology is well described but discussion section is comparatively less to explain the results.

Reviewer #11: The author substantially revised the manuscript and adequately addressed the queries raised during the reviewing process. It seems to me that the revised version can be accepted in this journal.

Reviewer #12: (No Response)

7. PLOS authors have the option to publish the peer review history of their article (what does this mean?). If published, this will include your full peer review and any attached files.

Reviewer #1: No

Reviewer #3: No

Reviewer #4: No

Reviewer #5: **Yes: **Mukhtar Ansari

Reviewer #6: **Yes: **Abdullah Salah Alanazi

Reviewer #7: No

Reviewer #8: No

Reviewer #11: **Yes: **Dr. AHM Khurshid Alam

Reviewer #12: No

---

## [Author Response · Author response to Decision Letter 1]

1 Jan 2022

Thank you for giving as the opportunity to submit a revised draft of my manuscript titled “Perception of physicians towards electronic prescription system and associated factors at resource limited setting” we appreciate the time and effort that you and the reviewers have dedicated to provide valuable feedback on the manuscript. We are grateful to the reviewers for their insightful comments on our paper. We have been able to incorporate changes to reflect most of the suggestions provided by the reviewers. We have highlighted the changes within the manuscript. 

Here is a point-by-point response to the reviewers’ comments and concerns. 

 Comments from reviewer 1

Comment 1: The manuscript needs extensive English editing

Response: Thank for pointing out such a valuable comment, we have edited the manuscript accordingly.

 Comments from Reviewer 3

Comment 1: They should have sent the questionnaires to 450-500 questionnaires in order to get a valid sample size of at least 384. As such, the inappropriate sample size should be acknowledged as one of the limitations of this study.

Response: Thanks for the comment based on your comment we have included this limitation under conclusion section of the manuscript line 10-11. 

 comments by reviewer 5

Comment 1: Keywords are still not in alphabetical order

Response: we arranged the key words in alphabetical order.

Comment 2: I am still not convinced with accessing current prescription practice through the question "do you have legible handwriting?

Response: Your point here is valid, but as we have mentioned in the method section, we used validated tools from the previous study conducted in India.

Comment 3: Proportional allocation of physician/hospital need to elaborate in the manuscript as well. 

Response: Thank you for raising such important issue that we have missed; we have incorporated this in the method section of the revised document. 

Comment 4: Ethical approval number and date need to be specified in the manuscript.

Response: we have included this revision under ethical consideration section.

 In addition to the above revisions all grammatical and language usage style comment raised by the reviewers were addressed accordingly.

---

## [Editor Report · Decision Letter 2]

5 Jan 2022

Perception of physicians towards electronic prescription system and associated factors at resource limited setting 2021: Cross sectional study

PONE-D-21-14066R2

Dear Dr. Hailiye,

We’re pleased to inform you that your manuscript has been judged scientifically suitable for publication and will be formally accepted for publication once it meets all outstanding technical requirements.

Kind regards,

Dylan A Mordaunt, MB ChB, MPH, MHLM, FRACP, FAIDH

Academic Editor

PLOS ONE

Additional Editor Comments (optional):

Thank you for your resubmission. This now meets the criteria for publication.
---

## [Editor Report · Acceptance letter]

10 Mar 2022

PONE-D-21-14066R2 

*Perception of physicians towards electronic prescription system and associated factors at resource limited setting 2021: Cross sectional study*

Dear Dr. Hailiye Teferi:

I'm pleased to inform you that your manuscript has been deemed suitable for publication in PLOS ONE. Congratulations! Your manuscript is now with our production department. 

Kind regards, 

on behalf of

Dr. Dylan A Mordaunt 

Academic Editor

PLOS ONE